# A Systematic Review of the Transthoracic Impedance during Cardiac Defibrillation

**DOI:** 10.3390/s22072808

**Published:** 2022-04-06

**Authors:** Yasmine Heyer, Daniela Baumgartner, Christian Baumgartner

**Affiliations:** 1Institute of Health Care Engineering with European Testing Center of Medical Devices, Graz University of Technology, A-8010 Graz, Austria; 2Clinical Division of Pediatric Cardiology, Department of Pediatrics and Adolescent Medicine, Medical University of Graz, A-8036 Graz, Austria; daniela.baumgartner@medunigraz.at

**Keywords:** transthoracic impedance, defibrillation, influencing factors, impedance distribution

## Abstract

For cardiac defibrillator testing and design purposes, the range and limits of the human TTI is of high interest. Potential influencing factors regarding the electronic configurations, the electrode/tissue interface and patient characteristics were identified and analyzed. A literature survey based on 71 selected articles was used to review and assess human TTI and the influencing factors found. The human TTI extended from 12 to 212 Ω in the literature selected. Excluding outliers and pediatric measurements, the mean TTI recordings ranged from 51 to 112 Ω with an average TTI of 76.7 Ω under normal distribution. The wide range of human impedance can be attributed to 12 different influencing factors, including shock waveforms and protocols, coupling devices, electrode size and pressure, electrode position, patient age, gender, body dimensions, respiration and lung volume, blood hemoglobin saturation and different pathologies. The coupling device, electrode size and electrode pressure have the greatest influence on TTI.

## 1. Introduction

The most effective method of treating ventricular tachyarrhythmias is rapid termination by an electric countershock, known as defibrillation. The magnitude and the distribution of the applied current play critical roles in defibrillation and are directly related to the TTI across the capacitor discharge of the defibrillator [1,2]. By definition, the TTI indicates the resistance of the thorax to the flow of current and is measured to check whether the defibrillation electrodes are correctly attached to the patient’s thorax. This affects the current amplitude and energy and thus the success rate of defibrillation. A smaller TTI means that less energy must be delivered to the body to achieve adequate current through the heart. A smaller transthoracic current flow could help prevent myocardial cell damage, necrosis and skin burns [3]. Various experiments have shown that of the total applied current, only 4% flows through the myocardium [4]. 82% are shunted off by the skeletal muscle layer and 14% by the lungs [1]. These observations highlight the need to analyze and determine the various factors that influence TTI in order to optimize the energy dosage and the current paths, and thus defibrillation outcomes. Some factors are constant and cannot be changed, such as the particular tissue impedance or chest volume [5]. However, other factors are determined by the applicant or the hardware of the defibrillation device and can be changed and adjusted to individual needs.

The range and limits of human TTI are of great interest for cardiac defibrillator testing and development. Therefore, the aim of this review is to determine the range of possible TTI in humans, the maximum and minimum TTI, in order to define the limits of patient impedance. In addition, TTI and its relationship to various influencing factors are examined in more detail, including the shock waveforms and different shock protocols, the coupling devices, electrode size and pressure, the placement of electrodes, patient age, gender, body dimensions, respiration and lung volume, the blood hemoglobin saturation and the impact of various pathologies.

## 2. Materials and Methods

This review was performed according to the Preferred Reporting Items for Systematic Reviews and Meta-Analysis (PRISMA) statement [6].

### 2.1. Data Sources and Search Strategies

Keywords were defined based on the aim of this review. The first two main keywords, KW1 = “transthoracic impedance” and KW2 = “defibrillation”, were combined for an initial search, and then a third keyword (KW3) related to specific characteristics was added, and the search was repeated. The databases NCBI (PubMed Central) and Google Scholar were searched with all keywords and their combinations shown in Figure 1.

### 2.2. Selection of Studies

For this systematic review, the relevant literature on TTI measurements and influencing factors between 1970 and 2021 was considered. Inclusion and exclusion criteria are summarized in Table 1. TTI measurements recorded for both defibrillation and cardioversion were included in this survey because both therapies treat patients with arrhythmias by delivering energy to the thorax. The only difference is that the shocks are synchronized with the QRS complex in cardioversion and are not so in defibrillation. The TTI is not affected by this difference. For a meta-analysis of human TTI additional criteria were defined (see also Table 1).

Figure 2 shows the systematic approach of literature identification, screening and inclusion. A total of 144 articles from database search and 28 articles from other methods were pre-selected for their relevance to the aim of the review using the abstracts of the papers. The aforementioned criteria (Table 1) formed the basis of paper selection. Citations were searched and reviewed using the bibliographies of the previously selected literature.

Articles deemed relevant were screened for eligibility, and the full text was carefully analyzed and categorized into different topic groups (TTI meta-analysis and review of the various TTI influencing factors) using RefWorks and Mendeley as reference management software. Finally, a total of n=71 articles were included in this review.

### 2.3. Data Extraction and Analysis

The analysis of the data was divided into two parts:Meta-analysis of all human TTI data with the aim of defining the range of possible TTI in humans including the maximum and minimum values of TTI;A detailed analysis of TTI influencing factors with the aim of assessing the relevance of each influencing factor on TTI.

For the meta-analysis, all human impedance measurements collected from the selected literature were included in Table A1 along with the measurement details, which were selected based on the investigated influencing factors (i.e., patient age, gender, weight, electrode type and placement, etc). Because of the large differences in the recording of TTI in the different studies, the technical details of the measurements and data on arrhythmia of the patients were documented as well to better compare the data. In some studies, more than one series of measurements was performed and all recordings with different configurations were included. Basic statistical methods were used for statistical analysis. Impedance results are reported as mean ± SD in Ω. Matlab^®^ (MathWorks) was used to create statistical plots for the presentation and analysis of the data.

The literature findings on different factors influencing TTI were reported, discussed qualitatively and compared for each influencing factor individually. Finally, an overall trend of how the factor alters TTI was given, along with a conclusion.

## 3. Results and Discussion

### 3.1. TTI an Ohmic Resistance

In various studies, the applied voltage and transversing current were recorded and virtually no phase difference could be detected [1,7,8]. This was found to be true for different waveforms, including damped sine wave and biphasic shocks. In terms of clinical defibrillation, resistance and impedance are equivalent. From these observations, it follows that TTI can be considered purely resistive according to Ohm’s Law [1].

The layer between the electrode and the body surface could act as a capacitor, but at the high current densities that occur during defibrillation, the capacitances are negligible [1].

### 3.2. Recording Methods for TTI

The recording of shock TTI values for research purposes is problematic for ethical reasons. It is not possible to apply high currents to healthy human hearts to record voltage and calculate the TTI. Only patients undergoing defibrillation or cardioversion can thus be monitored by modified defibrillators and a shock impedance at which the actual countershock can be recorded. This is, of course, easier for cardioversion because this procedure can be scheduled.

The other possibility is the estimation of the high-frequency impedance, which is measured before or without actual shock. This is a test-pulse method, where a weak current (generated by approximately 5 V at 20–30 kHz [9]) is passed through the thorax, and by applying fitting methods, the TTI can be predicted. The accuracy and validity of this method has been evaluated in different studies [9,10,11]. In some measurements of these studies, individual differences between test pulse and on shock impedances of 15–17% were found. However, since this was not the case for all measurements and a correlation coefficient of r = 0.986 was obtained between the test pulse impedance at 50 kHz and the actual shock impedance, the high-frequency measurements are sufficiently accurate to be considered and accepted as TTI in this analysis [8].

Table A1 in the Appendix A summarizes the different measurement methods as additional information to the TTI recordings.

### 3.3. Statistical Analysis of TTI Recordings

The mean TTI, maximum and minimum TTI (minimal, mean, maximal value and SD; not weighted to the number of subjects in the study) were calculated for the entire data set summarized in Table A1. The results are shown in Table 2.

In the following section, the statistical results summarized in Table 2 are discussed and references to Table A1 are provided. The documented measurements are numbered and for simplicity, No. # refers to the corresponding measurement in Table A1.

#### 3.3.1. Analysis of Mean TTI Recordings

The maximum observed mean TTI was 162 Ω, which was measured by Bissing and Kerber (see No. 79) [9]. In this case, the TTI was recorded from hairy patients, and poor electrode/tissue coupling could explain the exceptionally high mean TTI. A high value of 148 Ω was reported by Samson et al. in a pediatric study (see No. 98) [12]. Such high impedance values in pediatric defibrillation are common and are also found in other pediatric measurements (see No. 90–99) because of the small electrodes. All other mean TTI values are below 112 Ω, resulting in an averaged mean TTI value of 77.8 Ω. This value is within the range of 70–80 Ω given as the average TTI value for adults in the American Heart Association (AHA) Guidelines [13].

The minimum mean TTI was found to be 36 Ω by Caterine et al. (see No. 59) [14]. These measurements were made with gel between the electrodes in the AA position. This explains the very low impedances caused by the low impedance path created by the gel smear. However, there is no guarantee that a patient will receive sufficient current to the myocardium under these circumstances. All other mean TTI values showed at least 49 Ω.

The mean TTI SD values across all studies ranged from 3–36 Ω. The highest SD value with paddles was measured by Kerber et al. [15]. The paddle pressure can significantly alter TTI (see Section 3.4.5) and could be the cause of the high SD value. The smallest TTI SD value of 3 Ω was measured by Bissing and Kerber (see No. 81) [9]. All patients in this study had similar characteristics in that they were all shaved, hairy males. Two different studies (see Nos. 82–89) that examined different hairy patients, and coupling agents also showed very small TTI SDs of 4.0–6.4 Ω. These observations suggest that using the same coupling minimizes the SD values of TTI SD, which is surprising with respect to hirsutism. Nevertheless, the influence of the electrode/tissue interface appears to be a dominant factor of TTI. The average TTI SD was 16.4 Ω, which again shows that impedance varies greatly due to all the different influencing factors.

The highest and lowest mean TTI were not recorded in many studies (*n* = 32). The lowest TTI was measured by Kerber et al. with 12 Ω (see No. 3) [15]. Maximum low values of 74 Ω were reported in pediatric TTI examinations of infants aged 6 weeks to 9 months (see No. 92), indicating that body measurements alone are not determinant of TTI. In this study, small electrodes were used, and TTI decreased to a normal range after the use of adult paddles. The size of the electrode seems to outweigh the influence of body size and weight. The highest TTI of 212 Ω (see No. 90) was documented in the same study by Atkins et al. [16].

#### 3.3.2. Analysis of Adult TTI Data

Most of the extreme values discussed before have been attributed to pediatric measurements. However, the focus of this review is not on children and therefore pediatric data were excluded for the following histogram and boxplot representations. In addition, experiments on gel smears (Nos. 59 and 60), extremely hairy patients (No. 79) and without coupling gel (No. 89) were excluded because these measurement conditions represent extreme situations.

A mean average value of 76.7 Ω was calculated for the TTI in adults, which is 1.1 Ω smaller than the mean average value for the TTI including pediatric data. This is due to the high impedances caused by pediatric electrodes.

Figure 3 demonstrates the mean TTI characteristics of adults found in the literature, where the mean TTIs show almost normal distribution. The highest frequency of mean TTI is in the range of 73–80 Ω, which complies well with the median value given in the boxplots (75.0 Ω) as shown in Figure 4.

The boxplot in Figure 4, as a second visual representation of the TTI data, complements and reinforces the histogram plots. The spectrum of 25th and 75th percentiles of mean TTI found in the literature ranges from approximately 67.1–83.5 Ω. The median value of 75.0 Ω (min 51 Ω, max 112 Ω) is only slightly below the middle of the percentiles. The SD boxplot is evenly split, with a central median at 16.5 Ω.

### 3.4. Analysis of Influencing Factors

Further analysis of TTI in this review was narrowed down to a systematic view of the components that influence TTI. The different influencing factors found in the literature review are shown and categorized in Figure 5. This analysis also includes data from animal studies.

In the following sections, the 12 influencing factors are described and discussed in detail.

#### 3.4.1. Influence of Waveforms

The influence of the shock waveforms applied by the defibrillator and the resulting effects on TTI are presented and discussed in this section.

In a randomized study with damped monophasic sine waves (n=77) and rectilinear biphasic shocks (n=88) conducted by Mittal et al., the mean TTI for biphasic shocks was 76 ± 17 Ω and 78 ± 16 Ω for the monophasic waveform. The rectilinear biphasic waveform was found to be beneficial in patients with high impedance because it was insensitive to changes in TTI [17].

Similar results were observed by Page et al. TTI was higher in patients treated with monophasic shocks in whom cardioversion had failed in at least one shock than in patients immediately converted to regular heart rhythm (p=0.02). No difference in TTI (p=0.94) was observed in biphasic patients [18].

In the study by Niemann et al., mono- and biphasic waveforms applied after 90 s vf were compared. Here, a marginally significant decrease in TTI (p<0.05) was observed with biphasic waveforms applied to pigs [19]. Although the cause of this change could be attributed to the waveform, further explanations were not possible.

It can be concluded that the biphasic waveform is less sensitive to changes in TTI. The greater defibrillation efficiency of the biphasic waveform could not be attributed to the decrease in TTI. Further studies are needed in this area to determine the influence of the waveform on TTI and the resulting effects.

#### 3.4.2. Influence of Serial Shocks

As early as 1975, Pantridge et al. described the use of successive shocks when the first shock had failed. A decrease in impedance was thought to explain the success of repeated shocks with the same energy [20]. Nevertheless, the effect of repeated shocks on TTI remains controversial. While five studies support Pantridge et al. and the idea of a decrease in TTI [15,21,22,23,24], three other studies failed to confirm a change in TTI [11,25,26].

In the study by Kerber et al., the TTI was 52 ± 19 Ω at the initial shock and decreased to 48 ± 16 Ω (p<0.01) at the second shock. The changes in TTI were small and, according to the authors, not of clinical significance [15]. In a larger clinical study with *n* = 307 patients conducted by Walsh et al., a decrease in TTI with successive shocks was observed in both the AA and AP positions. The greatest decrease was noted between the first two shocks, with 4.3 ± 1.6 Ω in AA and 3.9 ± 4.5 Ω in AP positions [21]. However, a relatively high SD values raise concerns about the significance of the reduction and the overall impact of repeated shocks in TTI.

A small, but significant decrease in TTI of 2.7% with each subsequent shock (n=37) was found in the study conducted by Deakin et al. This could be due to the inflammatory response of the human body to the electric-current-induced damage [22]. The observations by Sirna et al. of erythematous rings on the chest of patients after removal of the electrodes support this theory. Deakin et al. also suggested that ionization phenomena might affect TTI. The impedance decreases when the energy used for defibrillation is increased. This effect could accumulate with successive shocks. The results for monophasic and biphasic defibrillation were comparable in this study [22].

In a study conducted only one year later, in 2009, Fumagalli et al. were able to confirm the above results. Fluid changes were thought to be the cause of the TTI changes because inflammation may increase vascular permeability and thus reduce impedance. The authors considered plausible the possibility of using multiple shocks with the same energy to achieve more efficient results with less tissue damage [24].

A total of 863 records of automated external defibrillators for the effect of multiple shocks were examined by Walker et al., but the authors came to a very different conclusion. In most cases, impedance remained unchanged, and in almost as many cases, the impedance decreased (n=149) or even increased (n=124) after the first shock. Notably, shocks of the same energy resulted in only minimal changes in TTI [11]. The major difference of this study compared with previous ones is that data from outpatient cardiac arrests were analyzed. Although the AHA Guidelines recommend the use of stacked shocks [27], the authors disagree with this and suggest moving away from this technique [11].

Monophasic and biphasic waveforms were tested on a swine model by Niemann et al. Again, repeated shocks did not alter TTI for both waveforms [25]. In Koster et al. only a small increase in TTI from 103 ± 24 Ω to 106 ± 23 Ω was documented. Here, impedance was measured before each shock using a high-frequency, low-level signal [26].

The reviewed articles could not give a clear answer on the influence of stacked shocks on TTI. Several studies reported decreases rather than stable TTI. All observed decreases were small with a high SD, so the significance of these results remains questionable.

#### 3.4.3. Influence of Coupling Devices

Different electrodes, pastes and gels have been developed and tested for use in order to optimize coupling of the electrodes with the body.

Using self-adhesive electrode pads, a TTI of 75 ± 21 Ω was measured by Kerber et al., which was slightly higher than with standard-size electrode paddles in comparable previous studies (67 ± 36 Ω) [28]. Similar findings were reported by Deakin et al., where Hewlett Packard^®^ self-adhesive pads, PhysioControl^®^ paddles and Hewlett Packard^®^ paddles were compared, and all three electrode types yielded similar TTI results (62.9 Ω, 64.7 Ω and 68.3 Ω). Using the self-adhesive pads lead to slightly lower TTI values than the paddles. However, gel impedance was not recorded directly in this study [29].

Impedance was found to be significantly higher (p<0.0001) by Dodd et al. when electrode pads were used instead of paddles in AA and AP positions [30]. The reduction in TTI with paddles is probably mainly due to the higher force applied by the operator (see also Section 3.4.5). In addition, the surface area of the pads was 1 cm${}^{2}$ smaller than that of the paddles (see Section 3.4.4). All these factors can accumulate and lead to the observed reduction.

Various gels and pastes are used for better contact during defibrillation. Redux paste was chosen as a coupling agent in a pediatric study conducted by Atkins et al. with significantly reduced TTI of 13% (p<0.05) [16]. Consistent with this study, Sirna et al. also documented an increase in mean TTI of more than 100% without the use of a coupling agent from 65 ± 5 Ω to 160 ± 5 Ω (n=37, p<0.01). However, this study was performed with paddles and not pads as in the previous study [23].

Two different electrode pads (Littman Defib^®^ pads and Harco^®^ pads) without an additional coupling layer and with Redux^®^ paste (Hewlett-Packard) as a coupling gel were tested in dogs by Aylward et al. TTI was significantly lower with the Redux^®^ paste compared to the two pads without paste (by approximately 12 Ω with p<0.01) [31].

A human study by Caterine et al. examined TTI during gel smearing between two electrodes. A reduction of 38% (58.0 ± 10.3 Ω to 36.0 ± 7.6 Ω) was observed with smeared gel. The low impedance pathway created between the electrodes decreased the predicted transcardiac current ratio by 22%. This problem could be anticipated and compensated for by apex-anterior positioning.

As concluded by Caterine et al., gel applications and smearing had the greatest effect on TTI [14].

#### 3.4.4. Influence of Electrode Size

One of the first TTI studies was conducted by Geddes et al., where animals of different sizes where investigated. Larger electrodes were used for bigger animals to reduce the impedance and allow sufficient current to cross the thorax [32]. Several animal and human studies followed, and all consistently showed that increasing paddle or pad size reduced TTI [9,12,15,28,29,32,33,34].

A mean TTI of 112.2 ± 17.0 Ω for small self-adhesive electrocardiogram defibrillator pads with a diameter of 8 cm was measured in the study by Dalzell et al. For electrodes with 8 cm in apex and 12 cm parasternal position, a TTI of 92.3 ± 22.0 Ω was obtained. For both electrodes with a diameter of 12 cm, the TTI further decreased to 71.6 ± 14.0 Ω [33].

Comparable results were documented by Kerber et al. with 13 cm and 8.5 cm paddles and by Bissing and Kerber for different electrode areas of 192 cm^2^ and 164 cm^2^, respectively [9,15].

Electrode size as a factor influencing TTI is one of the most important issues in pediatric defibrillation. Pediatric electrodes have been developed to fit on the chest of children. These are much smaller and result in significantly higher TTI than adult electrodes (p<0.0001). Mean impedances of 148 ± 23 Ω were measured for pads with diameters of 4, 4.5 and 5.8 cm and only 49 ± 9 Ω for diameters of 8 and 10 cm [12]. A more detailed consideration of the influences of TTI in pediatric defibrillation can be found in Section 3.4.7.

It has been shown that TTI decreases with larger electrode diameter, which increases current flow and thus could improve the success rate of defibrillation. At the same time, however, the use of electrodes that are too large could cause the current to pass through the myocardial tissue by seeking other routes through the thorax [15]. This assumes that there is an optimal electrode size that allows for optimal transmyocardial current and, thus, the best defibrillation results [29,33]. Indeed, for a given electrode size, the intracardiac current is optimized for each individual. This was investigated by Hoyt et al. in a study of mongrel dogs [34].

#### 3.4.5. Influence of Electrode Pressure

The force applied to electrodes significantly reduced TTI, as demonstrated in several studies [5,15,23,35,36,37,38,39,40].

A 25% reduction in TTI with firm paddle contact pressure compared to light paddle contact pressure was measured by Kerber et al. in a study with dogs. Contact at the electrode or skin surface is improved by increasing the number of low-resistance electrical contact points. In this study, Redux^®^ paste was used as a coupling agent. Increased pressure could lead to a more homogeneous distribution of the contact paste and could thus also contribute to the reduction in TTI. Paddle pressure and paddle size were found to have an additive effect on TTI. A larger paddle size plus paddle pressure resulted in a 40% reduction in TTI [15].

These observations were confirmed by Sirna et al., who provided evidence from their data [23]. A similar human study was published by Deakin et al., where the paddles were placed anteroapically. TTI decreased exponentially with applied paddle force up to 12 kg, and 8 kgf was sufficiently high and physically achievable for 80% of defibrillator operators [35,36]. The reduction achieved with 8 kgf on the electrodes was due to improved skin contact and reduced lung volume from pressure on the thorax (see Section 3.4.10) [37].

In addition to the absolute force, the electrode size over which the force is distributed plays an important role in determining the TTI. For smaller electrodes, the force per unit area is higher than for larger electrodes applying the same force. For this reason, the pressure on the electrodes must also be considered with explicit reference to the size of the electrode area. Small electrodes are of greatest importance in pediatric defibrillation. In the study by Deakin et al., TTI was found to be optimally low when pediatric paddles (16 cm${}^{2}$) with only 3 kgf are used for the defibrillation of infants weighing less than 10 kg. With adult paddles (82 cm${}^{2}$), approximately 5 kgf is used for the defibrillation of children ≤10 kg (see Section 3.4.7) [5].

Not only the force on paddles but also the force applied to self-adhesive defibrillator pads can reduce TTI. This was observed in a study by Persse et al. in a swine model during countershock and by Cohen et al. in a study of active compression cardioversion in human [38,39]. Ramirez et al. also measured a significant reduction in TTI with pressure on the pads in the AL position but not in AP-positioned pads [36]. In a study by Sado et al., the paddle force even reduced the effect of hirsutism, which increases TTI. At 1 kgf, for example, the difference in the mean TTI between unshaved and shaved men was 13.2 Ω [40].

In summary, it can be said that increasing the electrode pressure is one of the most influential factors in reducing TTI. For adult defibrillation with electrode paddles, 8 kgf is recommended [35,37].

#### 3.4.6. Influence of Electrode Position

To clarify the influence of the electrode position on the TTI, the positions and their abbreviations are clearly defined in Figure 6.

Different AA positions, the apex-to-anterior (AA1) and parasternal-to-anterior position (AA2), were compared by Caterine et al., and almost the same TTI of 58.0 ± 10.3 Ω and 51.0 ± 10.9 Ω, respectively, was measured [14]. Another fact to consider when positioning both electrodes in anterior positions is the gel smearing that may occur between the two electrodes. This could create a low-resistance pathway, as described in Section 3.4.3.

A study by Deakin et al. examined the longitudinal and transversal placement of the apex rectangular paddle in AA1 positioning. For all paddle forces less than 12 kgf, a higher TTI was measured in the transverse position. The above observation is confirmed when paddle force is also considered. The difference in TTI was greatest at low paddle forces and decreased at high paddle forces [41]. Very similar TTI values for different anterior–posterior (AP) measurements were found by Garcia and Kerber. Apex-right infrascapular (AP1) yielded 76.8 ± 18.4 Ω, apex-left infrascapular (AP2) 72.1 ± 18.7 Ω, anterior-right infrascapular (AP3) 72.5 ± 19.4 Ω and anterior-left infrascapular (AP4) 71.6 ± 18.6 Ω mean TTI [42]. Interestingly, TTI does not differ in the posterior positions either.

Most of the studies reviewed compared different AA and AP placements. The data on AA and AP positions are compared in Table 3. From there, we gather that there is a tendency for AA positioning to yield a slightly higher TTI than AP positioning. Similar observations can be made with respect to the AL position relative to the AP position. A meta-analysis by Zhang et al. of 10 randomized clinical trials involving n=1281 patients concluded that TTI was higher in the AL group than in the AP group, confirming the above results [43].

In summary, TTI was lower in the AP position than in the AA and AL electrode positions, although in some studies this was not sufficient to be significant. Most authors agree that despite slight differences, pad positioning is not a critical determinant of TTI [21,42,44,45,46,47,48].

#### 3.4.7. Influence of Age

Since human physiognomy changes with age, it was hypothesized that TTI could be influenced by these changes.

Pediatric electrodes have been developed for defibrillation in children. These electrodes are smaller in diameter than adult electrodes. This results in higher impedance and reduces peak current output by up to 25% [49].

TTI was measured by Samson et al. in children with electrodes of different sizes (4 to 10 cm diameter) covered by pre-gelled stannous chloride pads. For diameters of 4, 4.5 and 5.8 cm, the mean TTI was 148 ± 23 Ω, compared with 49 ± 9 Ω for diameters of 8 and 10 cm [12]. It is noticeable that the TTI recorded using the smallest electrodes is twice as high as the value of 76.7 Ω measured in adults, demonstrating that the influence of electrode size is more dominant than the age-related changes of the body. For older children, larger electrodes can be used, taking arcing into account [49]. For children older than one year, electrodes with a diameter of 7 cm were recommended [12]. This is consistent with the findings of Atkins et al., who observed that TTI was reduced by 47% when adult electrodes (83 cm${}^{2}$) were used instead of pediatric paddles (21 cm${}^{2}$) [16].

In the study by Samson et al., it was found that children’s TTIs were higher than expected based on their weight and body measurements alone. The recommendation of $2\frac{J}{kg}$ shock energy may need to be reconsidered based on the TTI analysis and increased accordingly [50]. TTI decreased less with applied force when pediatric paddles were used, as demonstrated by Deakin et al. The use of adult paddles for children up to 8 years of age resulted in a maximum reduction in TTI of 39.4% with paddle force and for children from 9 to 17 years of age, in a maximum reduction of 26.3% [5].

Significant differences in TTI as a function of age (p<0.01) was found by Garcia and Kerber. TTI was higher with age in three electrode positions (anterior-apex, anterior-posterior and apex-posterior). Impedance measurements also correlated with the BSA. To exclusively examine the effect of age on TTI, measurements were divided into two groups with BSA < 1.8 m${}^{2}$ and >1.8 m${}^{2}$ (p>0.001). For both groups, TTI increased with age, as shown in Table 4.

The authors suggested that the increase in TTI could be due to body posture in age, with pathological symptoms such as kyphoscoliosis. This could lengthen the path between electrodes, leading to higher TTI [42]. In contrast, Fumagalli et al. found that age had no effect on TTI. However, this result can be explained by the fact that 75% of patients were older than 69 years [2].

There are major differences between pediatric and adult defibrillation. The use of the electrodes, the optimal paddle force, energy matching and serial shocks must be considered and adapted to the needs of children. Generalization is difficult because of the very different characteristics of children at different ages and developmental stages. Even though one study found TTI to be generally higher with age [42], literature regarding a correlation of age and TTI was too scarce and sample sizes too small to draw any meaningful conclusions on this behalf.

#### 3.4.8. Influence of Gender

The results regarding the relationship between TTI and gender were controversial, and no clear trend was found. The effects of sex-typical characteristics on TTI such as hairiness, pregnancy and placement of electrodes on the chest were investigated and are presented below.

In the study by Caterine et al., the TTI was recorded for males and females in different application scenarios, showing that the impedance was lower for females without gel smear (see Table 5). These differences in the results for males and females could be due to the different distribution of subcutaneous tissue in the sexes. The placement of electrodes on the female breast was avoided [14].

A higher TTI in women than in men was measured by Fumagalli et al. A total of 80 patients (45% women) were studied using 78 cm${}^{2}$ self-adhesive defibrillator pads, with TTI measured after the first shock. Using electrical impedance tomography, forearm conductivity was measured higher in male subjects. These results were attributed to the different structure of adipose tissue between the sexes and to the statistically higher fat increase in women with age. Estrogen-dependent modulations in tissue may also be important for TTI when comparing the sexes [2]. This explanation was confirmed by Roh et al., denoting a higher mean TTI in women than in men (77 ± 15 Ω) vs. 63 ± 11 Ω, p<0.001) [51]. Three further studies by Garcia and Kerber, Koster et al. and Dalzell and Adgey found no significant difference between the sexes in TTI [26,42,52].

Chest pilosity is more common in men than in women and is therefore regarded as a feature to be considered as a possible influencing factor for male patients. Krasteva et al. concluded that TTI was slightly higher (p>0.05) when measured in men with pilosity due to poorer skin contact [48].

Furthermore, Bissing and Kerber compared male subjects who were shaved and unshaved, and reported a 35% decrease in mean TTI after shaving. In addition, hairy versus non-hairy patients yielded a significantly lower mean TTI (59 Ω, p<0.01) for the patients without pilosity. In this study, exceptionally high mean TTI values were documented for hairy patients, suggesting that shaving before defibrillation is necessary to ensure adequate electrode contact, especially in patients with severe hairiness [9]. The discrepancy between the two studies in terms of hairiness may be due to the difficulty in determining the degree of hirsutism. A third study by Sado et al. attempted to address this issue by defining the hair mass in grams. A regression analysis was performed, demonstrating a decrease in TTI after shaving as a function of hair mass [40].

In the study by Nanson et al., TTI was recorded before and after a pregnancy. Mean impedance was 91.3 ± 15.8 Ω within 2 weeks before anticipated delivery and 91.6 ± 11.8 Ω 6–8 weeks after delivery, when physiological status had normalized. Despite significant physiological changes, such as an increase in total body water of 6–8 liters and a 40–50% increase in blood volume, measurements on n=45 women showed that impedance remained unchanged [53].

Another characteristic that must be considered in women is the positioning of the electrodes directly on the female breast. TTI was measured by Pagan-Carlo et al. by placing the apex electrode on the breast, below and lateral to the breast. The mean TTIs were reported to be 95 ± 25 Ω, 84 ± 17 Ω and 83 ± 20 Ω, respectively. Positioning directly on the breast resulted in a significantly higher TTI (by 14% with p<0.01) than in the other two positions. In the large-breasted group with a higher brassiere size was the increase even greater (at 16%) [54].

A tendency for TTI to correlate with gender is difficult to discern because of the inconsistency of results in different studies. This could lead to the conclusion that gender does not affect TTI in general, but further studies are needed to confirm this. However, characteristic differences between the sexes (pilosity and the female breast) would need to be investigated on a case-by-case basis.

#### 3.4.9. Influence of Body Dimensions and Mass

The relationship between body measurements and TTI was not as readily apparent in the various studies as expected. The measurement and classification of the body mass required definition. Different indicators such as weight, chest size, BSA and BMI were used in the different studies, making a comparison difficult.

A study by Page et al. with n=203 subjects found that patients in whom initial shock cardioversion failed had a higher TTI of 83 ± 16 Ω vs. 71 ± 19 Ω (p=0.0007), which correlated with a higher weight of 93 ± 23 kg vs. 80 ± 17 kg [18]. From reports such as this, the debate arises about how much energy should be applied during defibrillation of severely obese patients. If energy needs to be adjusted upward, an impact of weight on TTI is implied.

A statistically significant but small relation between body weight and TTI was documented by Deakin et al. In this study n=80 children aged 10 weeks to 17 years were examined. Using adult paddles for children of less than 10 kg weight, a significant relation was found at 5.0 kgf paddle force (p<0.0001). The authors note that this relationship is not significant enough to be of clinical relevance [5].

Another major change in body weight can be observed during pregnancy. Nanson et al. found that TTI was unchanged before and after child delivery. The mean TTI of all participants was 81.6 Ω at birth and 70.5 Ω after delivery [53]. The lack of detailed patient data unfortunately limits the conclusions that can be drawn from the study. Another problem is the use of ’weight’ as a determinant of body mass. By merely measuring the weight of the patients, no information is given about the constitution of the body. Body mass relevant to the defibrillation process may not correlate with weight at all, as is evident in pregnancy. Overall, additional body fluids in the abdomen and the fetus account for the additional weight and therefore do not affect TTI during defibrillation.

This suggests the use of a different body parameter. In the study by Kerber et al., the influence of chest width and body weight on TTI was investigated. TTI was found to be weakly correlated with weight and more strongly correlated with chest width. Different paddle sizes were used, and a weak correlation with weight was found only for the standard size paddles (r=0.45,p<0.05) [15]. The effects of paddle size changes on TTI must be of greater influence on TTI, so that the minimal influence of weight is not significant. Of all body measures considered in this study, TTI correlated most strongly with chest width [15].

Chest size and weight were also measured in a study by Krasteva et al. The mean weight of the patients (n=86) was 75.9 ± 15.1 kg and the chest circumference was 103.4 ± 10.7 cm (range 86 to 130 cm). A weak correlation (r<0.5, p<0.05) between both parameters and TTI was found [48].

Another commonly used body characteristic is the BMI. A direct relationship between TTI and BMI was found in four studies [2,42,51,55]. In a bivariate, age-adjusted analysis with n=80 patients, a linear increase with r=0.574 and p=0.001 was found [2]. Wan et al. confirmed the positive correlation of TTI with BMI (r=0.33,p<0.01, n=623) using a wearable cardioverter-defibrillator [55]. A multivariate logistic regression analysis came to the same conclusion (B = 1.598, p<0.001). The authors of this study explain the increase in TTI by adipose tissue [51]. In the study by Garcia and Kerber, the BSA ranged from 1.4–3.0 m${}^{2}$ (mean 2.2 ± 0.9 m^2^), and a linear relationship was found (r=0.55, p<0.001). The authors conclude that increasing distance between the electrodes must increase TTI directly [42].

Overall, the studies reviewed show little, if any, rise in TTI with increasing body dimensions. The difficulty of using meaningful body characteristics and the wide range of influencing factors on TTI make a reliable analysis very difficult. In contrast to the hypothesis, the influence of body dimensions seems to be small. Nonetheless, research in the field of defibrillation of severely obese patients needs to continue to ensure that successful defibrillation is accessible to all.

#### 3.4.10. Influence of Respiration and Lung Volume

Lung volume has been found to increase TTI in seven studies [23,36,37,56,57,58,59].

In the study by Sirna et al. a 9% lower mean TTI was observed during end-expiration compared to end-inspiration (50 ± 3 Ω vs. 55 ± 3 Ω) [23]. In another study, the onset of expiration coincides with the maximum drop in the ventilation TTI curve [59]. These results were confirmed by Deakin et al., where higher lung volumes appear to result in longer current paths, leading to greater TTI [56].

High PEEP was found to additionally increase TTI in a linear relationship. For example, in the study by Deakin et al., a PEEP of 20 cm H_2_O increased TTI by 6%. At high PEEP levels, the change in TTI can be substantial [56].

The relationship between lung volume and paddle force was also investigated by Deakin et al. Fixed lung volume and variable lung volume tests were performed. It was found that TTI was higher in the fixed lung volume group for all measured paddle forces. The effect of lung volume was greatest at approximately 2 kgf of paddle force. Beyond 2 kgf , the decrease in TTI from baseline with increasing paddle force was dominated by the electrical contact between the paddle and skin. Only 16% of the total decrease in TTI could be attributed to changes in lung volume [37].

A different aspect on lung volume changes was investigated by Kim et al. In this study TTI was measured during various body positions and activities. TTI was affected by vital capacity in the upright, stationary position. In the supine position, an increase in TTI was found with decreasing end-inspiratory volume. This was not expected by the authors and was not consistent with the previous results presented above. A possible explanation could be that other factors (mechanical and/or physiological) dominate the influence on TTI. When TTI was examined during exercise, impedance was found to remain unchanged during the end-inspiratory increase in lung volume. These findings were unexpected and may again be due to other factors [57].

Different gas compositions and their influence on TTI were investigated in another study by Deakin et al. A total of 10 patients were tested with breathing air, 100% O_2_, 30% O_2_ + 70% N_2_O and 30% O_2_ + 70% He. No significant differences in TTI were found. Because of the lower conductivity of gases compared with tissue, it is likely that the current paths around the gas volumes in the lungs pass through blood and tissues. This explains the unchanged TTI results for different gas compositions in the lungs [58].

In summary, full lung volume may lead to a slight increase in TTI in humans, but it is not a major determinant of TTI. To compensate for influences of respiration, TTI measurements can be made at end-expiratory time points. Greater stability of TTI with respiratory changes was reported by Ramirez et al. in subjects with high BMI [36].

#### 3.4.11. Influence of Blood Hemoglobin Saturation

The authors of three different studies agreed that Hb O${}_{2}$ saturation is directly proportional to TTI [2,24,60].

Using multivariate linear regression analysis, Fumagalli et al. found that an increase of $1\frac{g}{dl}$ raised TTI by 1.9 ± 0.6 Ω (p=0.004) [2]. This change in impedance can be attributed to the electrical properties of the blood, which are affected by Hb concentration. It is believed that plasma is the conductive property, and the erythrocytes change the viscosity. It is also thought that endothelial cells and red blood cells serve as an insulating layer around the capillary wall [2]. These and other observations may explain the relationship between TTI and Hb saturation.

An even smaller increase in TTI of 0.2 ± 0.1 Ω was also found by Fumagalli et al. (p=0.0392). A bivariate linear regression model was used to obtain the slope of TTI versus Hb O${}_{2}$ saturation [24]. An increase in TTI was seen in all (n=222) but five patients. Higher saturation was generally associated with higher TTI change. In this study, 9.5% of the patients were diagnosed with COPD. COPD leads to a reduction in muscle mass, increasing the proportion of adipose tissue. In addition, lung tissue appears less vascularized. This resulted in higher TTI compared to the other patients. The effect of Hb O${}_{2}$ saturation on TTI may be partially related to COPD and the increase in TTI associated with COPD symptoms [24].

A decrease in Hb O${}_{2}$ saturation may be caused by hypoxia, in which a part of tissue is insufficiently oxygenated. In an animal study by Wojtczak, it was found that cytoplasmic resistance can be affected by different ion concentrations after prolonged hypoxia. The main reason for the change in internal longitudinal resistance was found to be an increase in impedance at intercellular junctions [60].

It can be concluded that higher Hb O${}_{2}$ saturation results in a small increase in TTI. The influence of Hb O${}_{2}$ saturation is likely to be of clinical interest only in combination with pathological symptoms, providing atypical Hb O${}_{2}$ saturation values.

#### 3.4.12. Influences of Pathologies

Several pathologies were found to influence TTI outcomes. Pulmonary edema, hypothermia, sternotomy, CHF and COPD are reviewed in the following section for their effects on TTI .

In a study by Fein et al., TTI measurements were performed for n=27 normal subjects and 33 patients of 2 groups of pulmonary edema of different severity. The TTI of patients without edema and with moderate edema were very similarly distributed and hardly distinguishable from each other. A decrease in TTI of at least 5% correlated directly with radiographic evidence of severe pulmonary edema. The increase in fluid in the tissues and lungs increases conductivity and thus decreases resistance [61]. This is consistent with the findings of Peacock et al., where a significant decrease (p<0.0002) in TTI was found for patients with cardiomegaly or abnormal pulmonary fluid [62].

The aim of both studies was to predict pulmonary edema, and therefore, the setup was somewhat different from the measurements for defibrillation. In the experiment by Fein et al., a constant alternating current was applied with two electrodes, and the voltage drop was recorded with two others to calculate impedance [61]. In the second study, the IQ monitor from Renaissance Technology, Inc., was used to collect TTI data [62]. Nevertheless, the results can be applied to TTI during defibrillation, and lower TTI values can be expected in severe pulmonary edema.

The measurement of TTI during induced hypothermia in human patients is ethically problematic due to the undesirable side effects of greatly reduced body temperature. For this reason, transferable animal experiments in pigs and dogs were carried out.

In the study by Rhee et al., severe hypothermia (down to 30 °C) was found to facilitate defibrillation because of altered myocardial properties. Often, improved shock success can be attributed to higher peak currents, resulting from lower resistance. Unexpectedly, the TTI results were slightly higher (p<0.0001) in severe hypothermia than in normothermia [63]. These findings are confirmed by Savino et al., who found a significant increase (p<0.05) in TTI under hypothermic conditions in 22 mongrel dogs, 14 of which were hypothermic with a mean central venous temperature of 27 °C. The mean difference in TTI between the 2 groups was only 3 Ω [64]. Note that metabolic demand, energy demand and oxygen consumption are reduced during hypothermia, and all these changes could explain the effect on TTI [63].

Sternotomy is a major surgical intervention into the mediastinum. Kerber et al. reported that this surgical procedure greatly reduces TTI (p<0.01), with changes persisting after wound healing. The changes were most prominent in the apex-right parasternal and left parasternal-right infrascapular position and less pronounced in the lateral-lateral position, where the current path does not cross the incision path [46].

During wound healing, tissue edema, inflammation, pleural effusions and hyperemia are common body responses. The increase in blood and extracellular fluids results in improved current conduction and thus decreased resistance. Long-term decreased TTI is thought to be due to scarring and adhesions that alter the properties of tissue [46].

In a clinical study with n=222 subjects conducted by Fumagalli et al., 9.5% of them diagnosed with COPD showed a lower reduction in TTI with successive shocks compared with the other patients (p=0.033). COPD alters body characteristics by reducing muscular mass, increasing adipose tissue and decreasing vascularized lung tissue, which could be the reason for the reduction in TTI [24].

Increased thoracic water content in patients with congestive heart failure (CHF) could also explain a reduced TTI. In the study by Fumagalli et al., a bivariate, age-adjusted analysis was performed, which showed a mean reduction of 5.3 ± 2.0 Ω with statistical significance (p=0.009) [2].

Several (patho)physiological factors can affect conductance and thus TTI, especially when parts in the thorax are affected. The more advanced the disease, the greater the changes that affect TTI. In some pathologies, such as pulmonary edema, TTI has even been studied as a possible indicator for diagnostic purposes.

### 3.5. Summary of Influencing Factors and Clinical Consequences

All influencing factors found in the literature review are summarized in Table 6.

The factors are categorized in low, medium and high influence on TTI. In addition, a brief summary of the conclusiveness of the categorization based on the detailed analysis and discussion of the influencing factors in Section 3.4 is provided.

From a clinical point of view, we can conclude that the most influential factors on TTI were the coupling device, electrode size and electrode pressure. In this context, the most important parameters to reduce TTI and thus improve the defibrillation success rate are increasing the electrode pressure and applying the gel correctly to achieve optimal conductivity between the electrode and the skin. Although the tissue between the electrodes accounts for most of the impedance, the coupling at the electrode/tissue interface is thus responsible for the largest changes in TTI. Furthermore, larger electrode diameter increases current flow as expected, but electrodes that are too large could cause the current to not pass through the myocardial tissue optimally. For a given energy setting of the defibrillator, a too high TTI in turn reduces current flow and may impair the ability of the electrical shocks to defibrillate, whereas a biphasic shock waveform is less sensitive to changes in TTI in general. There is also a major difference between defibrillation in children and adults. Reduced electrode size and optimal paddle force, energy matching and serial shocks must be considered and adapted to the needs of children. No clear trend was found in the influence of gender, but little increase in TTI was observed with increasing body dimensions, especially with chest width. This may require adjustments to electrode configurations and energy settings. A full lung volume, however, may lead to a slight increase in TTI, but is not a major determinant of TTI. To compensate for influences of respiration, it is recommended that defibrillation be performed at the end of expiration to further reduce TTI. Pathophysiological factors such as changes in tissue composition, pulmonary edema, sternotomy and others may affect conductance and thus TTI and should be considered during defibrillation when possible in an emergency situation.

## 4. Conclusions

The transthoracic impedance (TTI) is a major determinant of the transthoracic current flow and must be controlled during cardiac defibrillation to increase the success rate of defibrillation. In this systematic review, we analyzed the range and limits of the TTI studied and reviewed 12 different factors influencing this parameter. The major finding of the meta-analysis was that the TTI in humans ranges between 12 and 212 Ω in the reviewed literature, also including pediatric TTI data. This range is even wider than previously expected. However, the mean TTI values range from 51 to 112 Ω when the outliers and pediatric measurements are excluded. From these data showing normal distribution, a mean TTI of 76.7 Ω (median 75.0 Ω) was calculated, confirming the mean TTI of an adult human, which is between 70 and 80 Ω, as stated in the American Heart Association (AHA) Guidelines [13].

The large number of influencing factors demonstrates the complexity to analyze TTI. In order to filter out a specific influencing factor, all other factors would have to be kept stable, which is not always possible. Many studies also lack detailed information (see Table A1), which somewhat limited the analysis of the data.

## Figures and Tables

**Figure 1 sensors-22-02808-f001:**
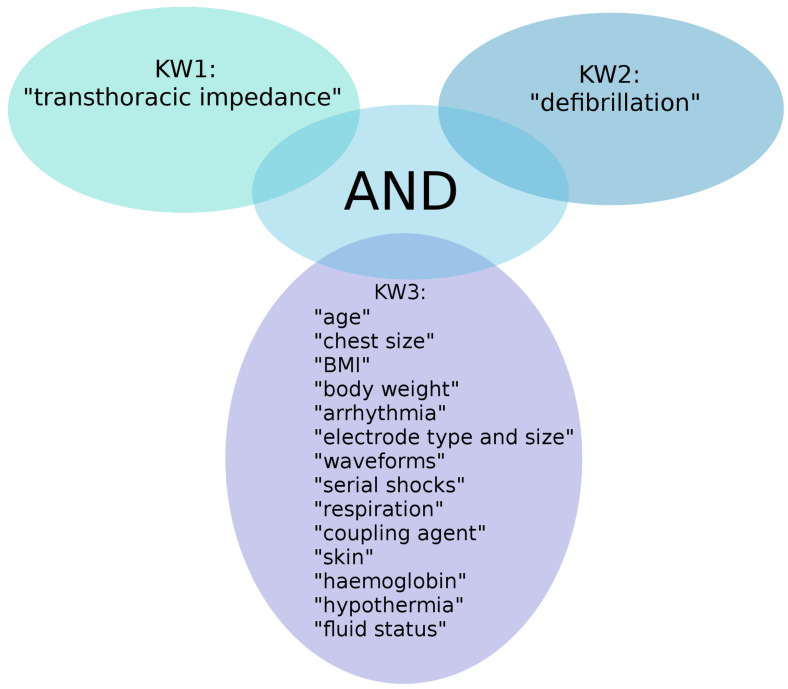
Keyword combinations which were used for the literature search.

**Figure 2 sensors-22-02808-f002:**
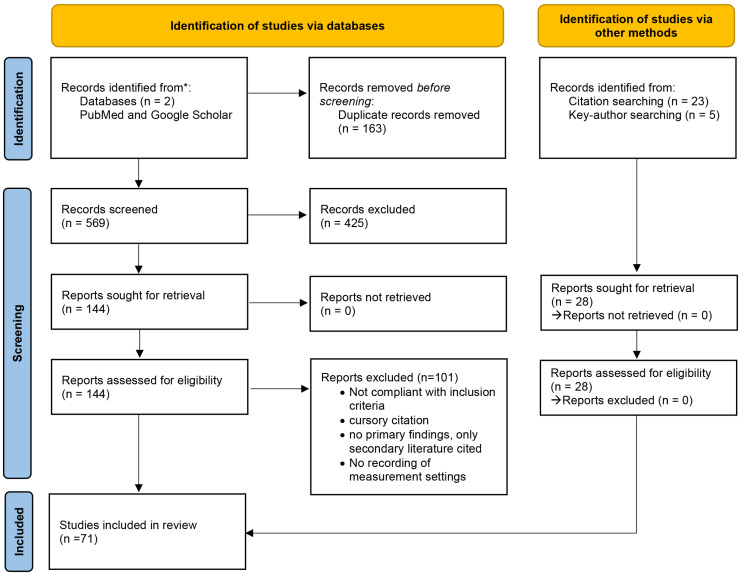
PRISMA 2020 flow diagram for updated systematic reviews which included searches of databases and other sources. See Page et al., 2021 [6]. For more information, visit: http://www.prisma-statement.org/ (accessed on 15 February 2022 ) [6].

**Figure 3 sensors-22-02808-f003:**
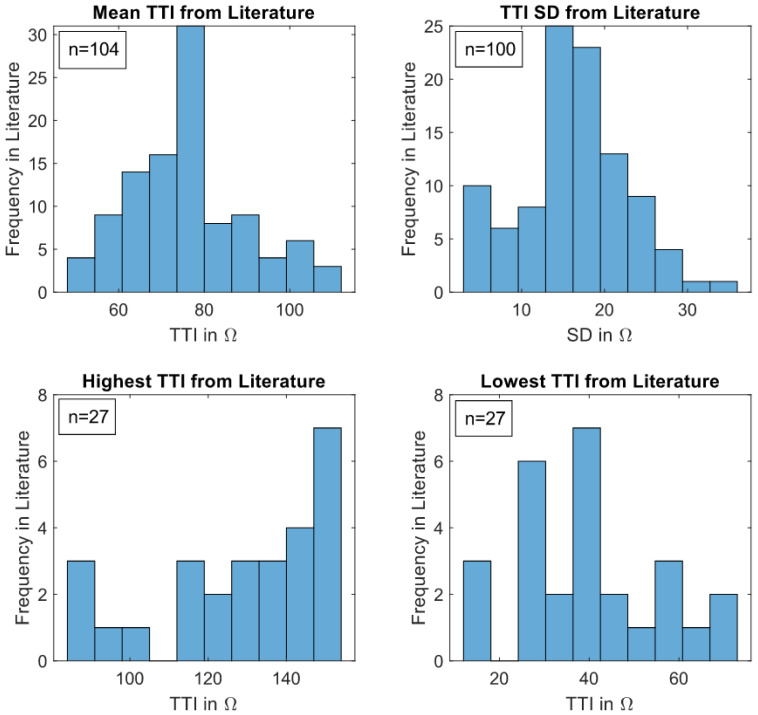
TTI data from Table A1 shown as histograms. In each subgraph, the number of trials used for histogram analysis is denoted by ‘*n*’.

**Figure 4 sensors-22-02808-f004:**
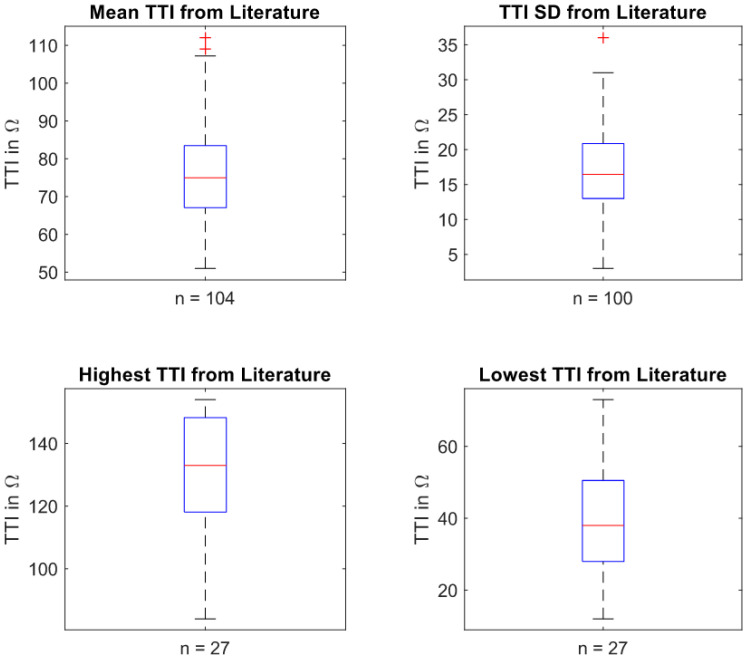
TTI data from Table A1 shown as boxplots. In each subgraph, the number of trials used for box plot analysis is denoted by ‘*n*’ below the graphs. The red line in the box indicates the median value. The upper and lower box boundaries represent the 25th and 75th percentiles, respectively. The most extreme data points are represented by the whiskers, excluding outliers. The latter are marked by a ‘+’ symbol.

**Figure 5 sensors-22-02808-f005:**
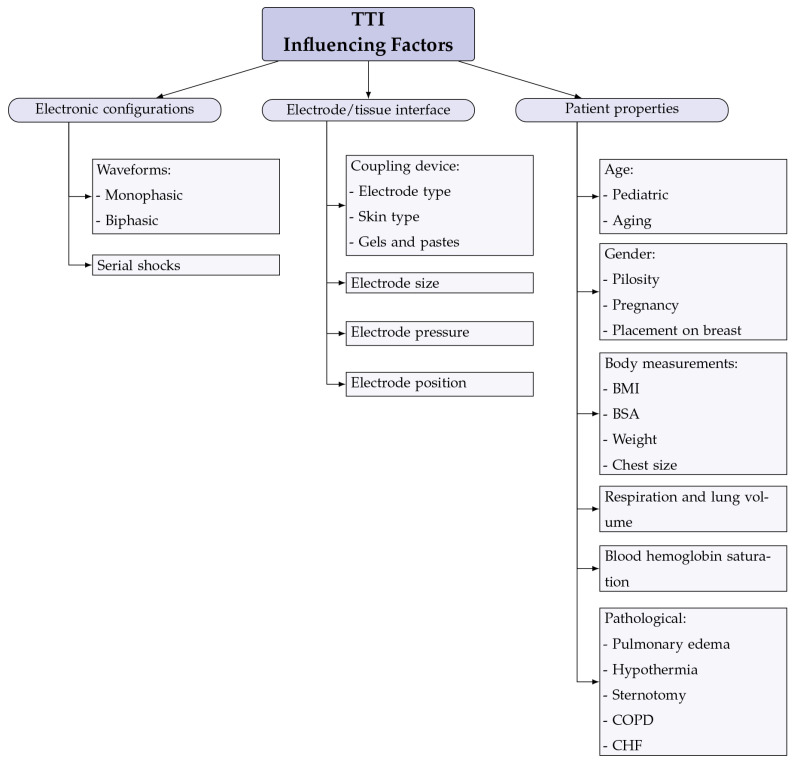
All TTI influencing factors found in the literature review and categorization into the different impedance determining components.

**Figure 6 sensors-22-02808-f006:**
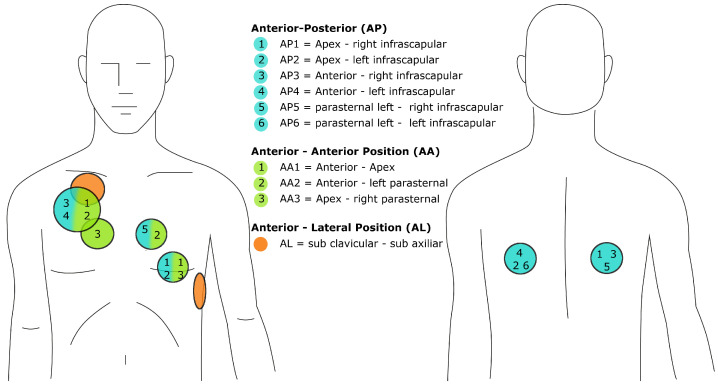
Positioning of the electrodes: AP positions (blue), AA positions (green) and AL positions (orange) each consist of two electrodes. The exact placement is indicated by the numbers and colors. For further reference, the additional abbreviations (AP1, AP2, etc.) are used in the text.

**Table 1 sensors-22-02808-t001:** Inclusion and exclusion criteria for the analysis of TTI influencing factors and meta-analysis of human TTI measurements.

	Exclusion Criteria	Inclusion Criteria
Reported Outcomes	No mention of TTI	Human and animal TTI measurements and analysis of factors influencing TTI
Exposure of Interest	TTI without connection to defibrillation	Defibrillation and cardioversion
	Implanted device-based methods	Electrode and paddle TTI measurements
	Relationship between tidal volume and TTI amplitude	
	Transthoracic impedance cardiography	
Setting	Impedance tracking in real time	Outpatient and inpatient hospital reports
Date	Papers published before 1970	Papers published 1970–2021
Language	Non-English	English
Additional criteria for meta-analysis of human TTI measurements	Animal measurements	Human TTI measurements only
	Crucial data missing (mean, highest and lowest)	Documentation of at least mean, highest or lowest TTI
	Case studies	Minimum of 5 subjects

**Table 2 sensors-22-02808-t002:** Statistical analysis of mean TTI, maximum and minimum TTI of all collected data.

Statistical Calculation	TTI in Ω			
	Minimum	Mean	Maximum	SD
Mean TTI	41.0	77.8	127.8	16.4
Maximum TTI	74.0	162.0	212.0	36.0
Minimum TTI	12.0	36.0	84.0	3.0
sample size (n)	32	118	32	100

**Table 3 sensors-22-02808-t003:** Different AA vs. AP TTI measurements: A comparison of the literature. The symbol AP without specification was denoted when no further information on AP position was available.

Literature	AA/AL Placement	AP Placement	TTI in AA Position Mean(± SD) in Ω	Mean TTI in AP Position Mean (± SD) in Ω	*n* for AA	*n* for AP
[44]	AA1	AP	73.8±16.8	65.5±14.5	31	39
[45]	AA1	AP	75.4±13	66.7±16	23	57
[42]	AA1	AP1	82.09±24.6	71.29±23.5	20	20
[46]	AA3	AP5	77±18	72±20	17	17
[47]	AA	AP	77.5±18.4	73.7±18.7	45	45
[21]	AA1	AP4	90±21	85±18	150	157
[48]	AL	AP2	107.2±22.3	96.6±19.2	86	86
[30]	AA1	AP6	92.6±16.99	92.1±23.3	21	21
[30]	AA1	AP6	75.8±14.1	66.5±13.9	21	21
[36]	AL	AP	81.4±17.6	70.9±16.6	11	11

**Table 4 sensors-22-02808-t004:** TTI mean ± SD (Ω) in relation to age (in years) and BSA (in m${}^{2}$) with n=20 [42].

TTI	Age ≤ 42	Age > 42
BSA < 1.8	55.3±9.0	86.6±13.7
BSA > 1.8	64.2±19.6	93.1±21.9

**Table 5 sensors-22-02808-t005:** Mean ± SD (Ω) of TTI at rest under different conditions determined by electrode position and gel application techniques in men (n=5) and women (n=5) [14].

Electrode Position	Gel Application Technique	Male Mean TTI ± SD (Ω)	Female Mean TTI ± SD (Ω)
Apex-to-anterior	Paddles only	65±9.3	54±4.0
Parasternal-to-anterior	Paddles only	55±8.8	47±6.3
Parasternal-to-anterior	Smeared	32±4.3	40±4.9
Apex-to-anterior	Smeared	54±6.8	48±3.6

**Table 6 sensors-22-02808-t006:** Factors influencing TTI found from the literature research are categorized according to the degree of influence (low, medium and high). An increase (↑) or decrease (↓) in TTI is represented by the respective symbol.

Factors	Section	Influence	Change of TTI	Conclusion
Waveforms	Section 3.4.1	low	no change	The biphasic waveform was found to be less sensitive to changes in TTI. The reduction in TTI for the biphasic waveform compared to the monophasic waveform was minimal.
Serial shocks	Section 3.4.2	low	↓ with shocks	Not all studies are congruent. More studies reported decrease rather than stable TTI. All recorded decreases were small and SD usually high.
Coupling device	Section 3.4.3	high	↓ for good coupling	Correct coupling (with gels/pastes and mechanical coupling of electrodes) at the electrode/tissue layer is a crucial factor for the TTI.
Electrode size	Section 3.4.4	high	↓ with electrode size	All studies are congruent. Of particular significance in pediatric defibrillation with very small electrodes.
Electrode pressure	Section 3.4.5	high	↓ with pressure	All studies are congruent. For adult defibrillation with paddles, 8 kgf is recommended.
Electrode position	Section 3.4.6	medium	↓ in AP	Overall, the TTI was lower in the AP position than in the AA and AL electrode positions (marginal in some studies). The TTI did not differ in the subgroups of the different AA positions nor in the AP positions. A higher TTI was found for the transverse position compared to the longitudinal position.
Age	Section 3.4.7	medium	↑ with age	Most of the differences are between adults and children, mainly because of body measurements that change with age. Only one study found TTI generally higher with age.
Gender	Section 3.4.8	low	↕	The results are inconsistent and no trends given. Characteristic differences between sexes can be considered: Breast hair should be removed to allow adequate coupling, on breast positioning of electrodes increases TTI, pregnancy does not affect TTI.
Body dimensions	Section 3.4.9	medium	↑ with body dimensions	Different indicators were used in the different studies: weight, chest size, BSA and BMI, making comparison difficult. A small, if any, increase in TTI with increasing body dimensions is indicated. Severely overweight should be considered.
Respiration and lung volume	Section 3.4.10	low	↑ with lung volume	Full lung volume may result in a slight increase in TTI. Measurements can be taken at end-expiratory time points to fully compensate for the of respiration.
Hemoglobin saturation	Section 3.4.11	low	↑ with Hb O_2_ saturation	Three studies are congruent: Higher Hb O_2_ saturation has a slightly increasing effect on TTI. Only of clinical interest in combination with pathological symptoms leading to very atypical Hb O_2_ saturation values.
Pathologies	Section 3.4.12	low	↕	Mainly minor changes of TTI with pathological symptoms were observed. Severe changes were only documented after sternotomy. In general, the more advanced the pathophysiological symptoms, the greater the change in TTI.

## Data Availability

Not applicable.

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
