# Peer review of "A Systematic Review of the Transthoracic Impedance during Cardiac Defibrillation"

_sensors, 2022, doi:10.3390/s22072808_

Round 1

Reviewer 1 Report

Summary

The authors present a very comprehensive and informative review of literature related to the transthoracic impedance of humans. There are a number of interesting subtopics within the review and I found it educational to read through. I feel that researchers (like myself) who are involved in cardiac arrest, ECG processing, and resuscitation research (among other areas) will find this a useful reference. The negative aspect of the review is that it is extremely long (11,000-12,000 words perhaps), although this is due in part to the fact that a number of topics are explored and explained. I believe this should be published and could serve as a good reference to future researchers, but that the word count should first be reduced if possible. One suggestion for reducing word could is to edit the wording to highlight fewer redundant details, and/or to perhaps remove majority of the text related to animal studies.

Specific points

  • I liked the overview in the introduction paragraphs.
  • With regards to the keywords for including a paper in the review, I am not certain that the bottom-middle category in the venn diagram would include most papers that are of interest according to the inclusion criteria in Table 1. Specifically, many studies about chest impedance in the out-of-hospital defibrillation setting may not include these keywords necessarily. The authors should consider expanding the keyword list to include not only “arrhythmia” but specific versions of this, such as “ventricular fibrillation”, “ventricular tachycardia”, and then perhaps keywords such as “cardiac arrest” and “resuscitation”. Or, the authors should explain why these additional keywords were not included or if it would have made a difference.
  • The averaged mean TTI is a nice overview value to present as a summary statistic in the review. Was this number weighted to reflect the different numbers of patients in the different studies?
  • Why did the authors include references the 2010 AHA guidelines? AHA updates guidelines every 5 years; was the relevant TTI value not included for 2020 or for 2015?
  • Did the authors consider the defibrillator model/brand and how it relates to impedance?
  • Is there enough information to provide a graph of pad area vs TTI, or did not enough papers supply this? I would be curious to see a plot of pad area versus TTI. Perhaps it’s out of the scope of the review too but I am curious.
  • The major point I would like to raise is that the review seems to have too many words. My rough estimate places the word count at around roughly 11,000-12,000. The MDPI author guidelines state that a review should be a minimum of 4000 words but it does not specify a maximum, so I suppose technically this is acceptable. However, the size makes the paper hard to read, and is due in part to a lot of detail being included. For example, just the text that describes the distribution of TTI plots (mean and SD plots of TTI across the studies) is perhaps 600 words. The level of detail could be reduced by editing the text for brevity. I understand that much of the detail is important, but perhaps some of it is redundant and could be streamlined. Please make an effort to reduce the number of words where possible, and report the original word count versus the reduced word count. Much of the detail is important but please make an effort to remove some extra wording where possible.
  • The text initially says that review is in regards to human TTI, but then there is discussion of animal TTI in the text. One option to keep the review more focused is to limit it only to human studies; this would help reduce word count as well. I know that some work went in to finding the animal TTI studies but since the study is focused on human, perhaps it’s best to just put a short mention to the sections where animal TTI papers are described (e. g. hypothermia) but reduce the detail included.
  • The grammar was good overall; I did not observe any obvious typos.
  • I will also point out that there were many interesting points that I was not aware of (e.g. blood oxygen is related to TTI)

Author Response

REVIEWER 1

We would like to thank the reviewer for his/her very valuable comments and recommendations on our review paper, which we have carefully considered in the revised manuscript. Below please find our detailed comments.

The authors present a very comprehensive and informative review of literature related to the transthoracic impedance of humans. There are a number of interesting subtopics within the review and I found it educational to read through. I feel that researchers (like myself) who are involved in cardiac arrest, ECG processing, and resuscitation research (among other areas) will find this a useful reference. The negative aspect of the review is that it is extremely long (11,000-12,000 words perhaps), although this is due in part to the fact that a number of topics are explored and explained. I believe this should be published and could serve as a good reference to future researchers, but that the word count should first be reduced if possible. One suggestion for reducing word could is to edit the wording to highlight fewer redundant details, and/or to perhaps remove majority of the text related to animal studies.

Specific points

  • I liked the overview in the introduction paragraphs.

We thank the reviewer for this positive comment!

  • With regards to the keywords for including a paper in the review, I am not certain that the bottom-middle category in the venn diagram would include most papers that are of interest according to the inclusion criteria in Table 1. Specifically, many studies about chest impedance in the out-of-hospital defibrillation setting may not include these keywords necessarily. The authors should consider expanding the keyword list to include not only “arrhythmia” but specific versions of this, such as “ventricular fibrillation”, “ventricular tachycardia”, and then perhaps keywords such as “cardiac arrest” and “resuscitation”. Or, the authors should explain why these additional keywords were not included or if it would have made a difference.

We thank the reviewer for this important comment. We decided to use the hypernym “arrhythmia” because it covers the broad spectrum of specific manifestations such as (sub)ventricular fibrillation, ventricular tachycardia, etc. and is usually used in the literature along with the term “arrhythmia”. We also agree that more specified keywords such as cardiac arrest or resuscitation would be useful, however we decided to use the two main keywords “transthoracic Impedance” and “defibrillation” for the initial DB query, those representing the key terms for quantitative meta-analysis on TTI (first part of this review). The combination with the third keyword was used to search for the broad range of influencing factors that make up the second part of this review article. However, we believe that only a very small number of articles have been lost, not using more specified keywords. Because of the enormous amount of work involved in creating this review article, we apologize not to repeat the queries again and continue to expand this work. We hope that this can be accepted by the reviewer. 

  • The averaged mean TTI is a nice overview value to present as a summary statistic in the review. Was this number weighted to reflect the different numbers of patients in the different studies?

We used the unweighted data considering studies with at least n=5 subjects (see Table 1: Additional criteria for meta-analysis of human TTI measurements). We agree that too many studies with less cases or even single case studies could influence the statistical data. Therefore, studies with <5 subjects were excluded for TTI meta-analysis. We addressed this issue in the text that the data presented are unweighted.

“The mean TTI, maximum and minimum TTI (minimal, mean, maximal value, and SD; not weighted to the number of subjects in the study) were calculated for the entire data set summarized in Table A1.”

As the aim of this study was “to determine the range of possible TTI in humans”, so we decided to present the mean as an overview of what is reported as the average in the literature, i.e., the unweighted mean. The weighted mean was calculated to be 80.1 Ω, which is only 2.3 Ω higher than the unweighted mean of 77.8 Ω. Due to this small difference, we decided not to open a new chapter to compare the weighted and unweighted values.

  • Why did the authors include references the 2010 AHA guidelines? AHA updates guidelines every 5 years; was the relevant TTI value not included for 2020 or for 2015?

Thank you very much for this comment. We have reviewed the current AHA guidelines again. The relevant TTI value was not reported in the 2015 and 2020 guidelines. Therefore, the values from 2010 is still valid according to the reference from Circulation, 2010.

  • Did the authors consider the defibrillator model/brand and how it relates to impedance?

All available defibrillator models/brands were listed in Table A1 in the Appendix. In the main text of the review, we could not systematically provide this information because this data was not fully available or discussed as an influencing factor in the reviewed literature.

  • Is there enough information to provide a graph of pad area vs TTI, or did not enough papers supply this? I would be curious to see a plot of pad area versus TTI. Perhaps it’s out of the scope of the review too but I am curious.

We again reviewed the literature and found that the information on pad area vs. TTI was inconsistent and sparse (only 28 of original 119 recorded measurements). First, in some papers the pad areas were not specified. Second, the main problem was that the other factors influencing the TTI could not be calculated out because this information was not sufficiently available. Because of these limitations and the length of the paper we did not address this topic in the revised manuscript. We agree that this topic would be interesting and a survey on this topic is still missing.

  • The major point I would like to raise is that the review seems to have too many words. My rough estimate places the word count at around roughly 11,000-12,000. The MDPI author guidelines state that a review should be a minimum of 4000 words but it does not specify a maximum, so I suppose technically this is acceptable. However, the size makes the paper hard to read, and is due in part to a lot of detail being included. For example, just the text that describes the distribution of TTI plots (mean and SD plots of TTI across the studies) is perhaps 600 words. The level of detail could be reduced by editing the text for brevity. I understand that much of the detail is important, but perhaps some of it is redundant and could be streamlined. Please make an effort to reduce the number of words where possible, and report the original word count versus the reduced word count. Much of the detail is important but please make an effort to remove some extra wording where possible.

We thank the reviewer for this important comment! We agree with the reviewer and shortened the whole manuscript accordingly (see marked version of the paper). The statistical part (meta-analysis) was shortened from 4.5 to 3 full pages (-30%) and the analysis of 12 (!) influencing factors from 15.5 to 14 pages (-10%). In total minus 3 printed pages. It should be mentioned that two other reviewers asked for additional information in the manuscript (e.g. extension of the Conclusion section, summarizing the clinical relevance of this review, almost one page). It was really hard work to make up for the loss of important information while shortening this article and adding new information.  In addition to this, we believe that the readability has now been significantly improved. And, there is no page limitation by the publisher of Sensors.

  • The text initially says that review is in regards to human TTI, but then there is discussion of animal TTI in the text. One option to keep the review more focused is to limit it only to human studies; this would help reduce word count as well. I know that some work went in to finding the animal TTI studies but since the study is focused on human, perhaps it’s best to just put a short mention to the sections where animal TTI papers are described (e. g. hypothermia) but reduce the detail included.

Thank you for this important information. It is correct that the meta-analysis was performed only on human TTI. However, animal studies were considered in addition for reviewing and analyzing the 12 influencing factors. Animal studies are important because some technical experiments could not be performed on humans for ethical reasons. However, we shortened the text throughout the manuscript, also related to animal studies. 

  • The grammar was good overall; I did not observe any obvious typos.

Thank you for this comment. Nevertheless, a final proofreading was carried out by a professional translation service.   

  • I will also point out that there were many interesting points that I was not aware of (e.g. blood oxygen is related to TTI)

The authors highly appreciate this comment. We are very pleased that the reviewer enjoyed this article.

We again thank the reviewer for his/her time and effort to review this article.

Reviewer 2 Report

Dear Editor,

Thank you for the opportunity to review this document.

The document is well written and very complete, maybe too much.

The document has an atypical mix structure with result and discussion together, this somewhat makes sense but is not usual.

The main limitation is that is way too long. Basically 24 pages of text to read (plus some tables and references). And it is 24 pages in the final publication format which means probably 50 pages in word with double spacing. The document lecture gives the impression of a thesis lecture that was cut and past in a document and submit as is.

It gives the impression that the authors copy/past the result part of the abstract each of the 71 paper.

Even for a review of literature this is too long. Most not to say all reader will quit before the end. I forced myself to read up to page 11 and realize I still have more then half the text to read and quit the detail reading with only flying through the document. Even as a motivated reviewer this is too long.

The data is super interesting, super complete, super detail, too much detail.

The document needs to be rework and synthesized.

The references are up to date.

INTRO

The authors should define TTI in the main text.

RESULTS AND DISCUSSION

"11% to−1.5%"

This is a weird way to present this. The authors should spell it differently.

"The TTI SD values of the studies ranged from 3 − 36Ω."

It is very unusual to present result like this, even If I understand why the authors are doing so, and should keep this. However, table 2 is unclear. When I read the table I thought that the authors were presenting result as mean (SD). While They are presenting SD as a separate set of info. The table needs to be change. Please see below the table section for some proposition.

"very small TTI SDs of 4 − 6.40Ω."

I do not know if the 0 is a typo? The authors might want to correct it  to 6.4Ω.

“16.18Ω”

For most result having only 1 or even no number after the dot would be enough and make the document more readable without losing any valuable data.. 16.18 vs 16.2 vs 16.

This should be done through out the text, per example "39.95 and 128.52Ω” could be change to 40-129.

" The highest and lowest TTI were not recorded in many studies, so the sample size was reduced from n = 118 to 32. "

I would recommend to remove the 32 from the table as it makes it very confusing to have a min and max lower then the mean.

" The boxplot in Figure 4, as a second visual representation of the TTI data, complements and reinforces the histogram plots. "

This document is too long. Presenting somewhat the same data should not happen. Too much info kills the info.

FIGURE

Fig2, after reports assessed for eligibility (n=28) the authors might want to add a delete duplicate from the main databases analysis

TABLE

Table 2: presenting the result with mean TTI Then SD TTI then max TTI then minimum TTI makes the table hard to read. I would advise the authors to do min, mean, max, SD or mean, min, max SD. As explain above I would clearly sperate mean and SD as some people are mostly going to look the figures and not the text (It will also help It table is cut paste in PowerPoint.).

Table 2: for the sample size, lowest and max are 32. With a mean of 118 (SD 110). This probably needs to be change as explain above..

Author Response

REVIEWER 2

We would like to thank the reviewer for his/her very valuable comments and recommendations on our review paper, which we have carefully considered in the revised manuscript. Below please find our detailed comments.

Dear Editor,

Thank you for the opportunity to review this document.

The document is well written and very complete, maybe too much.

The document has an atypical mix structure with result and discussion together, this somewhat makes sense but is not usual.

The main limitation is that is way too long. Basically 24 pages of text to read (plus some tables and references). And it is 24 pages in the final publication format which means probably 50 pages in word with double spacing. The document lecture gives the impression of a thesis lecture that was cut and past in a document and submit as is.

It gives the impression that the authors copy/past the result part of the abstract each of the 71 paper.

Even for a review of literature this is too long. Most not to say all reader will quit before the end. I forced myself to read up to page 11 and realize I still have more then half the text to read and quit the detail reading with only flying through the document. Even as a motivated reviewer this is too long.

We thank the reviewer for this important comment! We agree with the reviewer and shortened the whole manuscript accordingly (see marked version of the paper). The statistical part (meta-analysis) was shortened from 4.5 to 3 full pages (-30%) and the analysis of 12 (!) influencing factors from 15.5 to 14 pages (-10%). In total minus 3 printed pages. It should be mentioned that two other reviewers asked for additional information in the manuscript (e.g. extension of the Conclusion section, summarizing the clinical relevance of this review, almost one page). It was really hard work to make up for the loss of important information while shortening this article and adding new information. In addition to this, we believe that the readability has now been significantly improved. And, there is no page limitation by the publisher of Sensors.

The data is super interesting, super complete, super detail, too much detail.

The document needs to be rework and synthesized.

We thank the reviewer for this very positive comment.

The references are up to date.

Thank you.

INTRO

The authors should define TTI in the main text.

We thank the reviewer for this comment. We added a definition of the TTI in the introduction.

“By definition, the TTI indicates the resistance of the thorax to the flow of current and is measured to check whether the defibrillation electrodes are correctly attached to the patient's thorax. This affects the current amplitude and energy, and thus the success rate of defibrillation”.

RESULTS AND DISCUSSION

"11% to−1.5%"

This is a weird way to present this. The authors should spell it differently.

Thank you. This paragraph was shortened and in particular this sentence was removed as it is not essential.

"The TTI SD values of the studies ranged from 3 − 36Ω."

It is very unusual to present result like this, even If I understand why the authors are doing so, and should keep this. However, table 2 is unclear. When I read the table I thought that the authors were presenting result as mean (SD). While They are presenting SD as a separate set of info. The table needs to be change. Please see below the table section for some proposition.

As mentioned before, this paragraph was significantly shortened to improve readability. Table 2 was changed according to the reviewer’s recommendations.

"very small TTI SDs of 4 − 6.40Ω."

I do not know if the 0 is a typo? The authors might want to correct it  to 6.4Ω.

Thank you. Corrected.

“16.18Ω”

For most result having only 1 or even no number after the dot would be enough and make the document more readable without losing any valuable data.. 16.18 vs 16.2 vs 16.

This should be done through out the text, per example "39.95 and 128.52Ω” could be change to 40-129.

We have decided to use one digit after the decimal point.

" The highest and lowest TTI were not recorded in many studies, so the sample size was reduced from n = 118 to 32. "

I would recommend to remove the 32 from the table as it makes it very confusing to have a min and max lower then the mean.

We changed table 2 as recommended by the reviewer.

" The boxplot in Figure 4, as a second visual representation of the TTI data, complements and reinforces the histogram plots. "

This document is too long. Presenting somewhat the same data should not happen. Too much info kills the info.

We have significantly shortened the meta-analysis part (from 4.5 to 3 pages), in particular the explanations of the figures with the histograms and boxplots. A lot of detailed information has been removed. 

FIGURE

Fig2, after reports assessed for eligibility (n=28) the authors might want to add a delete duplicate from the main databases analysis

This review was performed exactly according to the PRISMA guidelines. The guidelines did not suggest additional tracking of the number of duplicates from the additional search and the main database analysis. The database search was performed first, followed by the additional search (citation search and key author search), and papers already included were not searched again. We can confirm that this step was performed for deleting possible duplicates from the main database analysis before literature identification. The results show that there were no duplicates after literature identification.

TABLE

Table 2: presenting the result with mean TTI Then SD TTI then max TTI then minimum TTI makes the table hard to read. I would advise the authors to do min, mean, max, SD or mean, min, max SD. As explain above I would clearly sperate mean and SD as some people are mostly going to look the figures and not the text (It will also help It table is cut paste in PowerPoint.).

Table 2: for the sample size, lowest and max are 32. With a mean of 118 (SD 110). This probably needs to be change as explain above..

We thank the reviewer for this comment. We changed the table structure accordingly.

Finally, the paper was now proofread by a professional translation service.

We again thank the reviewer for his/her time and effort to review this article.

Reviewer 3 Report

The study deals with an interesting topic which is not very well examined.

The review is well designed and clearly conducted. 
To my mind, the introduction would benefit from a short section discussing the parameter TTI as such to open the article to a broader readership. Furthermore, illustrating this topic an additional figure would be nice showing what exactly is measured with TTI and how it is done.

Furthermore, a more extensive discussion about implications in the practical sourrounding would be helpful discussing how this data could influece clinical practice.

Author Response

REVIEWER 3

We would like to thank the reviewer for his/her very valuable comments and recommendations on our review paper, which we have carefully considered in the revised manuscript. Below please find our detailed comments.

The study deals with an interesting topic which is not very well examined.

The review is well designed and clearly conducted. 
To my mind, the introduction would benefit from a short section discussing the parameter TTI as such to open the article to a broader readership.

We thank the reviewer for this comment. We added an explanation of the TTI in the introduction.

“By definition, the TTI indicates the resistance of the thorax to the flow of current and is measured to check whether the defibrillation electrodes are correctly attached to the patient's thorax. This affects the current amplitude and energy, and thus the success rate of defibrillation”.

Furthermore, illustrating this topic an additional figure would be nice showing what exactly is measured with TTI and how it is done.

Because reviewers 1 and 2 advised that the article be significantly shortened, we decided not to add an additional figure. However, as now additionally addressed in the introduction, TTI is a major determinant of transthoracic current flow in defibrillation. For a given energy setting, a high transthoracic impedance reduces current flow and may adversely affect the ability of electric shocks to accomplish defibrillation. Some defibrillators use the TI measurement function to adjust the energy of the defibrillation pulse. Changes in tissue composition due to redistribution and movement of fluids induce fluctuations in the TI. We address these issues, along with the reviewer's concluding remarks about the impact on clinical practice, in the Conclusions section.

Furthermore, a more extensive discussion about implications in the practical sourrounding would be helpful discussing how this data could influece clinical practice.

We thank the reviewer for this important comment. We revised the Conclusions section comprehensively and addressed the impact of findings on clinical practice in this section.

“For the clinical practice, we can summarize that the most influential factors on TTI were the coupling device, electrode size and electrode pressure. In this regard, the most important parameters to reduce TTI and thus improve the defibrillation success rate are to increase the electrode pressure and to apply the gel correctly to achieve optimal conductivity between the electrode and the skin. Although the tissue between the electrodes accounts for most of the impedance, the coupling at the electrode/tissue interface is thus responsible for the largest changes in TTI. Furthermore, larger electrode diameter increases current flow as expected, but too large electrodes could cause the current not to pass through the myocardial tissue optimally. A constant finding was that TTI is lowest at end of exspiration, underscoring the importance of both defibrillation and cardioversion at end-exspiration. For a given energy setting of the defibrillator, a too high TTI in turn reduces current flow and may impair the ability of the electrical shocks to defibrillate, whereas a biphasic shock waveform is less sensitive to changes in TTI in general.

There is also a major difference between defibrillation in children and adults. Reduced electrode size and optimal paddle force, energy matching, and serial shocks must be considered and adapted to the needs of children. No clear trend was found in the influence of gender, but little increase in TTI was observed with increasing body dimensions, especially with chest width. This requires adjustments to electrode configurations and energy settings. However, a full lung volume may lead to a slight increase in TTI, but is not a major determinant of TTI. To compensate for influences of respiration, it is recommended that defibrillation be performed at the end of expiration to further reduce TTI as mentioned before. Pathophysiological factors such as changes in tissue composition, pulmonary edema, sternotomy and others may affect conductance and thus TTI, and should be considered during defibrillation when possible in an emergency situation”.

Finally, the paper was now proofread by a professional translation service.

We again thank the reviewer for his/her time and effort to review this article.

Round 2

Reviewer 1 Report

Thank you for considering my suggestions. I will say that the text is still quite long but perhaps this is acceptable to the journal, given the nature of this particular review. I wonder if a short "table of contents" with hyperlinks might help this particular document, given its length and number of subtopics.

The only last comment I have for the current version is that the Conclusion section is too long and probably shouldn't contain a Table. Generally a Table should not be presented in the conclusion. Is it possible to reorganize the text to make it easier to follow the conclusion? For example, perhaps the Conclusion section should only contain the first paragraph of the current Conclusion. Then, instead, before the Conclusion section, the authors could insert a "Summary" section which contains the summary Table 6 and some of the conclusion paragraph 2-3 text? Something like this might help the organization of the article.

Author Response

REVIEWER 1

We would like to thank again the reviewer for his/her very valuable comments and recommendations, which we have carefully considered in the revised manuscript. Below please find our detailed comments.

Thank you for considering my suggestions. I will say that the text is still quite long but perhaps this is acceptable to the journal, given the nature of this particular review. I wonder if a short "table of contents" with hyperlinks might help this particular document, given its length and number of subtopics.

We thank the reviewer for this comment. In accordance with the recommendations of reviewer 2 to further shorten the manuscript, we have additionally reduced redundant information and shortened the article by more than one printed page.

The only last comment I have for the current version is that the Conclusion section is too long and probably shouldn't contain a Table. Generally a Table should not be presented in the conclusion. Is it possible to reorganize the text to make it easier to follow the conclusion? For example, perhaps the Conclusion section should only contain the first paragraph of the current Conclusion. Then, instead, before the Conclusion section, the authors could insert a "Summary" section which contains the summary Table 6 and some of the conclusion paragraph 2-3 text? Something like this might help the organization of the article.

We thank the reviewer for this comment which further improves the readability of the article. We have restructured this section of the manuscript and introduced a new subsection “3.5 Summary of influencing factors and clinical consequences”. Table 6 is now in this section and the clinical summary has also been moved to this section.

We again thank the reviewer for his/her time and effort to review this article.

Reviewer 2 Report

Dear Editor,

Thanks, you for the opportunity to review the revised and shorten version of this document.

The document has been shortened a lot which improved the readability of the information.

However, I still think the document is too long.

I understand it is a review of literature, I also understand that the journal has not set up a maximum number of words which I wound surprising. Per example, Resuscitation and the European journal of emergency medicine have a word limit of 4000 for review of literature. The New England journal of Medicine 2500. Those limits if they make life hard for the authors are also there to force the authors to do a very complex work of synthesis and writing.  And also, to help the reader.

One example among a lot of other:

“Dalzell et al. measured a mean TTI of 112.2 } 17.0Ω for small self-adhesive electrocardiogram defibrillator pads with a diameter of 8cm. For electrodes with 8cm in apex and 12cm parasternal position, results of 92.3}22.0Ω were obtained. Moreover, for both electrodes with a diameter of 12cm, the TTI decreased to 71.6 } 14.0Ω. [33]

Comparable results were documented by Kerber et al. with 13cm and 8.5cm paddles. The shock energy applied was approximately the same in both studies. [15]

Bissing and Kerber confirmed previous results and found that the mean TTI decreased from 134 } 9Ω to 97 } 12Ω (P = 0.07) for 191.5cm2 and 164cm2 electrodes, 333 respectively. [9]”

In this case the information of the second and third paragraph are very redundant.

Another example, for the next topic:

There is a great conclusion/synthesis to the topic:

“Considering all this, it can be said in summary that increasing the electrode pressure is one of the most influential factors in reducing TTI. For adult defibrillation with electrode paddles, 8kg f is recommended”

However, there is literally 35 lines of text. That content is interesting but still could be reduce. Sadly, this probably means rewriting a lot of it.

I do not think that the result of each study needs to be presented as the authors are doing.

The authors are presenting:

Study A shows …

Study B shows …

Study C shows …

Study D shows …

Study E shows …

Study F shows …

In this presentation, the authors do basically on paragraph per study.

I think the message is stronger and easier to read when presented as:

Studies A, B, C demonstrate that  …

While study D and E demonstrate …

Strangely study F is in favor of.

I want however to finish on being clear that I think the paper needs to be publish. The content is great and bring an amazing synthesis on a very complex topic. The document will give the opportunity to the cardiac arrest community to have a reference paper for this topic. There is a thin balance of balance for the length of review of literature

Author Response

REVIEWER 2

We would like to thank again the reviewer for his/her very valuable comments and recommendations, which we have accordingly considered in the revised manuscript. Below please find our detailed comments.

Dear Editor,

Thanks, you for the opportunity to review the revised and shorten version of this document.

The document has been shortened a lot which improved the readability of the information.

However, I still think the document is too long.

I understand it is a review of literature, I also understand that the journal has not set up a maximum number of words which I wound surprising. Per example, Resuscitation and the European journal of emergency medicine have a word limit of 4000 for review of literature. The New England journal of Medicine 2500. Those limits if they make life hard for the authors are also there to force the authors to do a very complex work of synthesis and writing.  And also, to help the reader.

We thank the reviewer for this comment. We further improved the readability of the article and removed redundant information. The article was also shortened by more than one printed page.

One example among a lot of other:

“Dalzell et al. measured a mean TTI of 112.2} 17.0Ω for small self-adhesive electrocardiogram defibrillator pads with a diameter of 8cm. For electrodes with 8cm in apex and 12cm parasternal position, results of 92.3}22.0Ω were obtained. Moreover, for both electrodes with a diameter of 12cm, the TTI decreased to 71.6} 14.0Ω. [33] Comparable results were documented by Kerber et al. with 13cm and 8.5cm paddles. The shock energy applied was approximately the same in both studies. [15]Bissing and Kerber confirmed previous results and found that the mean TTI decreased from 134 } 9Ω to 97 } 12Ω (P = 0.07) for 191.5cm2 and 164cm2 electrodes, 333 respectively. [9]”

In this case the information of the second and third paragraph are very redundant.

We thank for this comment. We removed redundancy and shortened this paragraph as follows: 

“A mean TTI of 112.2 ±17.0Ω for small self-adhesive electrocardiogram defibrillator pads with a diameter of 8cm was measured in the study by Dalzell et al.. For electrodes with 8cm in apex and 12cm parasternal position, a TTI of 92.3 ±22.0Ω was obtained. For both electrodes with a diameter of 12cm, the TTI further decreased to 71.6 ±14.0Ω. [33] Comparable results were documented by Kerber et al. with 13cm and 8.5cm paddles and by Bissing and Kerber for various electrode areas of 192cm2 and 164cm2, respectively. [15][9]”

Redundant information has now been removed throughout the entire manuscript.

Another example, for the next topic:

There is a great conclusion/synthesis to the topic:

“Considering all this, it can be said in summary that increasing the electrode pressure is one of the most influential factors in reducing TTI. For adult defibrillation with electrode paddles, 8kg f is recommended”

However, there is literally 35 lines of text. That content is interesting but still could be reduce. Sadly, this probably means rewriting a lot of it.

 We also shortened this section.

I do not think that the result of each study needs to be presented as the authors are doing.

The authors are presenting:

Study A shows …

Study B shows …

Study C shows …

Study D shows …

Study E shows …

Study F shows …

In this presentation, the authors do basically on paragraph per study.

I think the message is stronger and easier to read when presented as:

Studies A, B, C demonstrate that  …

While study D and E demonstrate …

Strangely study F is in favor of.

We thank the reviewer for repeating this recommendation. We have changed this concern throughout the manuscript. No paragraph now begins with “Study A shows…”

I want however to finish on being clear that I think the paper needs to be publish. The content is great and bring an amazing synthesis on a very complex topic. The document will give the opportunity to the cardiac arrest community to have a reference paper for this topic. There is a thin balance of balance for the length of review of literature

We again thank the reviewer for his/her positive summary of our work and appreciate his/her time and effort to review this article.
